# Design, Synthesis, and Use of Novel Photoaffinity Probes in Measuring the Serum Concentration of Glycogen Phosphorylase

**DOI:** 10.3390/molecules24040798

**Published:** 2019-02-22

**Authors:** Yuchao Zhang, Youde Wang, Zhiwei Yan, Chengjun Song, Guangxin Miao, Liying Zhang

**Affiliations:** 1Key Laboratory of Traditional Chinese Medicine Research and Development of Hebei Province, Institute of Traditional Chinese Medicine, Chengde Medical University, Chengde 067000, China; zyc9021@126.com (Y.Z.); wangyoude8686@126.com (Y.W.); cdyanzhiwei@163.com (Z.Y.); mgx8088@163.com (G.M.); 2Department of Human Anatomy, Chengde Medical University, Chengde 067000, China; songchengjun@126.com

**Keywords:** CP-320626, glycogen phosphorylase, acute myocardial infarction, semiquantitative protein electrophoretic mobility shift technique, photoaffinity probe

## Abstract

A procedure to measure the serum concentration of glycogen phosphorylase during acute myocardial infarction is presented. This method was based on the synthesis of photoaffinity probes, and used the semiquantitative protein electrophoretic mobility shift technique. Three novel photoaffinity probes bearing different secondary tags were synthesized. Their potency was evaluated in an enzyme inhibition assay against rabbit muscle glycogen phosphorylase a (RMGPa). The inhibitory activity of probe **1** was only 100-fold less potent than the mother compound CP-320626. The photoaffinity labeling experiments were also performed, and a protein with molecular weight (MW) of about 90–100 kDa, which was consistent with the MW of GP, was clearly labeled by probe **1**. A semiquantitative evaluation of the GP level in serum with probe **1** was also performed. The results showed that the protein band with a MW of about 90–100 kDa was tagged, and the concentration of the protein in serum was found to be between 25 and 50 ng/mL. Mass spectrometric analysis revealed that alpha-1,4 glucan phosphorylase (GPMM) was well-preserved in the bands.

## 1. Introduction

Early identification and confirmation of myocardial injury is essential for correct patient care and disposition decision in the emergency department. In this respect, glycogen phosphorylase (GP) has become an enzyme for early laboratory detection of ischemia based on its metabolic function [1]. Three GP isoenzymes are present in human tissue. These are named according to the tissue of their initial description: GPLL (liver), GPMM (muscle), and GPBB (brain). Heart muscle contains GPMM and GPBB in high concentrations. During myocardial ischemia, activation of GP results in an increase in glycogen degradation. Thus, GP is released from glycogen and then enters the bloodstream, which is believed to occur via the T-tubulus system [2]. It is considered the most sensitive marker for the diagnosis of acute myocardial infarction (AMI) within 4 h after the onset of chest pain [3]. It is also considered a useful marker for the detection of perioperational myocardial ischemia and early reperfusion [4]. In addition, also it is considered to be a specific marker for detection of perioperative myocardial injury in patients undergoing coronary artery bypass grafting [2].

Clinical detection of GP requires rapid and reliable test systems. Immunoassay methods and chemical approaches have been developed to determine plasma GP concentrations during hypoxia and ischemia injury in the heart [5,6]. Although these studies have been somewhat successful, they have not yet reached definitive conclusions to allow the use of these approaches in routine determinations. This is because most studies have only examined the measurement of GPBB, but not the efflux of GPBB and GPMM. Continuous efforts for the development of methods that can accurately detect and quantify the amounts of GP in plasma during acute myocardial infarction are becoming more and more pressing.

Here we describe the development of a photoaffinity labeling method for the estimation of GP. This method is based on the synthesis of photoaffinity probes and use of the semiquantitative protein electrophoretic mobility shift technique [7]. The probes were designed based on CP-320626 (Figure 1), a high-affinity inhibitor of GP [8]. X-ray crystallographic imaging of the complex of CP-320626 with GP showed that the 5-chloro-1*H*-indole-2-carboxamide moiety had appropriate interactions with GP at the dimer interface site. The 4-hydroxy-piperidyl moiety exhibited relatively few Van der Waals interactions with the protein [9]. Therefore, chemical modification of CP-320626 was attempted by introducing a spacer arm in the hydroxy moiety to create a viable protein (GP) scaffold. The benzophenone moiety was incorporated as the photochemically reactive group because of its excellent chemical stability in solvents and easy preparation [10]. While a biotin tag is ideal for affinity tagging purposes, its use is not optimal for higher throughput activity-based profiling due to the time and labor involved in producing western blots. Additionally, many cells and tissues contain endogenously biotinylated proteins that complicate analysis of biotinylated probe-based western blots [11]. To circumvent this shortcoming, we synthesized a fluorophore-tagged probe **1**, which possessed a dansyl group that exhibited relatively intense fluorescence in water [12]. For the fluorescent probe, we added a biotin group to the structure of probe 1, creating a dual function probe **2**, making it suitable for both affinity purification and fluorescence applications [13]. Moreover, in order to avoid the steric hindrance caused by the reporter tag, we designed another tag-free probe **3** that employed an azide handle for downstream conjugation of the reporter tag, via the click chemistry reaction after proteome labeling [14].

Copper-catalyzed azide-alkyne cycloaddition (CuAAC) is a widely utilized, reliable, and straightforward way for making covalent connections between building blocks containing various functional groups. It has been used in organic synthesis, medicinal chemistry, surface and polymer chemistry, and bioconjugation applications. In the present study, we designed and synthesized a series of activity-based probes based on CP-320626 with a CuAAC reaction (Figure 2).

## 2. Results and Discussion

### 2.1. Chemistry

Introduction of an azide moiety into the structure of CP-320626 is summarized in Scheme 1. Reaction of tetraethylene glycol (**4**) with *p*-toluenesulfonyl chloride (TsCl) in CH_2_Cl_2_ at 0 °C gave tosyl ester **5** in 60% yield. The hydroxyl group of **5** was protected in the usual manner, with dihydropyran catalyzed by PPTS to afford the THP ether **6** (71% yield). Treatment of **6** with *tert*-butyl 4-hydroxypiperidine-1-carboxylate using sodium hydride as base produced the desired ether derivative **7**. Subsequent deprotection with 3 M HCl afforded **8** in excellent yield, which in turn was reacted with Boc-l-4-fluorophe in the presence of HATU and DIPEA to deliver piperidineamide **9**. Removal of the *N*-Boc group from **9** with 3 M HCl furnished intermediate **10**, and was directly reacted with 5-chloroindole-2-carboxylic acid in the presence of HATU and Et_3_N, resulting in alcohol **11** in 69% yield. Treatment of **11** with methanesulfonyl chloride (MsCl) and Et_3_N afforded mesylate **12**, which was converted to intermediate **13** via a nucleophilic substitution reaction with sodium azide.

Click chemistry was handled as shown in Scheme 2. The alkyne **16** bearing a benzophenone photophore and a fluorescent tag was readily prepared in five steps starting from Lys(Boc)-OMe (**14**) and 4-benzoylbenzoic acid (**15**) [15]. The corresponding azide intermediate **13** and alkyne **16** were dissolved in CH_2_Cl_2_-H_2_O, followed by the addition of catalytic sodium ascorbate and CuSO_4_·5H_2_O, resulting in triazole probe **1** in 11% yield.

The synthesis of the final probe **2** is shown in Scheme 3. It took seven steps to prepare the dual-labeled alkyne **18** starting from *N*-Cbz-*N*′-Boc-l-lysine **17** [15]. It was similar to probe **1**, where the reaction of the aforementioned azide **13** with the dual-labeled alkyne **18** resulted in probe **2** in 12% yield.

The tag-free probe **3** was also prepared as shown in Scheme 4. According to the reported method, synthesis of the corresponding chloroacetyl compound **19** was based on Lys(Boc)-OMe (**14**) as well as 4-benzoylbenzoic acid (**15**) [15]. Then, probe **3** was synthesized by the CuI-catalyzed Huisgen (2 + 3) dipolar cycloaddition reaction (click chemistry), followed by a nucleophilic substitution reaction with sodium azide.

### 2.2. Enzyme Assay and SAR Analysis 

To understand the validity of the synthetic probes, the synthesized probes **1**–**3** were evaluated in an enzyme inhibition assay against rabbit muscle glycogen phosphorylase a (RMGPa). As described previously, the activity of RMGPa was measured through detecting the release of phosphate from glucose-1-phosphate in the direction of glycogen synthesis, based on the published method. Caffeine, a known allosteric GP inhibitor that shares the same binding site with CP-320626, was used as a positive control [16]. The bioassay results are summarized in Table 1. Probes **2** and **3** had no inhibitory activity against RMGPa. As expected, the steric interference was crucial for the inhibitory effect. In contrast, probe **1** inhibited RMGPa with an IC_50_ value of 13.20 μM, and was approximately 100-fold less potent than that of CP-320626. To our surprise, the incorporation of the azide moiety resulted in a diminished activity in the assay. Due to the lack of a binding mode of chemical probes with GP, it is impossible to estimate how the introduction of different functional elements affected the three-dimensional conformation of GP. We only confirm that there were significant differences in the regulation of the active state of the GP enzyme between probe **3,** containing the azide group, and probe **1** without the azide group.

### 2.3. Labelling of Serum Proteomes with Probe ***1***

Based on the results of the potency in the enzyme assay, probe **1** was selected and used throughout photolabeling studies (Figure 3). The serum proteomes prepared from rats with acute myocardial infarction were incubated with probe **1** at concentrations of 10, 3, and 1 mM, and 300, 100, and 30 nM, respectively. The mixture was exposed to UV light for 30 min, and then was imaged using the Tanon-6100 Chemiluminescent Imaging system. The label transfer analysis is shown in Figure 3. Interestingly, using the new photoprobe **1**, the protein band with a MW of about 90–100 kDa, which was consistent with the MW of GP, was clearly labelled. These results demonstrated that the synthesized probe **1** might be used to label the GP in serum.

### 2.4. Semi-Quantitative Analysis of Serum-Free Glycogen Phosphorylase (GP) Levels with Probe ***1***

An attempt to semi-quantitatively analyze the serum-free GP level after an acute myocardial infarction was explored. Serum blanks were added in increasing quantities of RMGPa (0, 12.5, 25, 50, and 100 ng/mL at a final concentration) in the presence of an excess of probe **1** (10 μM). In the same way, probe **1** was added to the serum in an acute myocardial infarction model in a rat. All mixtures were exposed to UV light for 30 min, and then were imaged using the Tanon-6100 Chemiluminescent Imaging system. The label results are presented in Figure 4. These data demonstrate that the synthesized probe **1** efficiently labeled the protein bands with a MW of about 90–100 kDa, which was consistent with the MW of GP. The standard curve method was applied to detect the concentration of the protein’s MW consistent with the MW of GP (Figure 4, Lane 2). Figure 5 shows the standard curve of the protein concentration. Corresponding to the standard curve, the concentration of the protein’s MW consistent with the MW of GP was found to be between 25 and 50 ng/mL. For a more detailed proteomic analysis, we then used the mass spectrometric analysis method to detect the protein bands with a MW of about 90–100 kDa. Several proteins were identified, and the results are summarized in Table 2. This analysis revealed that the GPMM was well-preserved. However, it was unavoidable that the gel had typical high-abundance proteins such as cytoskeletal protein, deoxynucleotidyltransferase terminal, interacting protein, Meiosis 1-associated protein, Cation-transporting ATPase, etc. Interestingly, a homologous protein, alpha-1,4 glucan phosphorylase, appeared in the search results in Table 2. The detailed results of the LC-MS/MS analysis are in Figure 6. The parent ion with the highest protein percent coverage (74%), alpha-1,4 glucan phosphorylase (GPMM), was selected for fragmentation and MS/MS analysis. The origin of sequence ions from the protein indicated that variable modifications of GPMM, such as the acetyl of protein N-term and oxidation (M), was presented.

## 3. Materials and Methods

### 3.1. Chemistry

All commercially available solvents and reagents were used without further purification. Melting points were uncorrected. NMR experiments were performed on Bruker Avance III 400 MHz and Bruker Fourier 300 MHz (Bruker Corporation, Billerica, MA, USA). The spectra were referenced internally according to residual solvent signals of TMS (δ = 0.00 ppm). Positive or negative ion LCMS data were obtained at 303 K by a quadrupole mass spectrometer Agilent LC/MSD 1200 Series (Agilent technologies Inc, Palo Alto, CA, USA) using a 50 × 4.6 mm (5 μm) ODS column. Reversed-phase-HPLC experiments were performed by flash welchrom C18 column (150 × 20 mm) chromatography (Agela Technologies, Tianjin, China). The NMR spectrum of compounds were shown in the Appendix A).

#### 3.1.1. Ethyl 2-{2-[2-(2-hydroxyethoxy)ethoxy]ethoxy}-4-methylbenzenesulfonate (**5**)

Under an ice bath, TsCl (40 mL, 7.93 g, 0.042 mol) was slowly added in groups to a mixture of tetraethylene glycol (9.0 mL, 0.05 mol) and Et_3_N (18.05 mL, 0.13 mol) in anhydrous CH_2_Cl_2_ (60 mL). The reaction mixture was stirred at 0 °C overnight. Then, the reaction mixture was diluted with CH_2_Cl_2_ and washed with water, 1 M aqueous HCl, and saturated NaHCO_3_ and brine, and then dried and evaporated over anhydrous Na_2_SO_4_. The residue was purified by column chromatography on silica gel [EtOAc-petroleum ether (35–90%)] to give the product (8.77 g, 60%) of colorless oil. HPLC analysis was as follows: 99.0%; ^1^H-NMR (CDCl_3_, 400 MHz) δ 7.80 (d, *J* = 8.2 Hz, 2H), 7.35 (d, *J* = 8.0 Hz, 2H), 4.17 (t, *J* = 4.8 Hz, 1H), 3.72–3.68 (m, 4H), 3.66–3.62 (m, 4H), 3.61–3.58 (m, 6H), 2.64 (s, 2H), and 2.45 (s, 3H); ^13^C-NMR (CDCl_3_, 100 MHz) δ 144.9, 132.9, 129.8, 128.0, 72.5, 70.7, 70.6, 70.4, 70.3, 69.3, 68.7, 61.7, and 21.6.

#### 3.1.2. Ethyl 2-[2-(2-{2-[(tetrahydro-2H-pyran-2-yl)oxy]ethoxy}ethoxy)ethoxy]4-methylbenzenesulfonate (**6**)

A mixture of Pyridinium 4-toluenesulfonate (PPTS, 0.73 g, 2.90 mmol) and 3,4-2*H*-dihydropyran (3.93 mL, 43.11 mmol) in CH_2_Cl_2_ (50 mL) was added to a solution of compound **5** (10.0 g, 28.74 mmol) in anhydrous CH_2_Cl_2_ (50 mL). The reaction mixture was stirred at room temperature for 5 h. Then, the reaction mixture was diluted with CH_2_Cl_2_ and washed with water and saturated brine, dried over anhydrous Na_2_SO_4_, and evaporated. The residue was purified by column chromatography on silica gel [EtOAc-petroleum ether (15–50%)] to give the product (8.83 g, 71%) of colorless oil. HPLC analysis was as follows: 90.1%; ^1^H-NMR (CDCl_3_, 400 MHz) δ 7.80 (d, *J* = 8.2 Hz, 2H), 7.34 (d, *J* = 8.0 Hz, 2H), 4.62 (t, *J* = 3.2 Hz, 1H), 4.18–4.15 (m, 2H), 3.87–3.83 (m, 1H), 3.72–3.47 (m, 15H), 2.45 (s, 3H), 1.86–1.78 (m, 1H), 1.75–1.68 (m, 1H), and 1.63–1.49 (m, 4H); ^13^C-NMR (CDCl_3_, 100 MHz) δ 144.8, 133.0, 129.8, 128.0, 99.0, 70.7, 70.64, 70.58, 70.53, 69.2, 68.7, 66.6, 62.2, 30.6, 25.4, 21.6, and 19.5.

#### 3.1.3. *Tert*-butyl 4-{2-[2-(2-{2-[(tetrahydro-2H-pyran-2-yl)oxy]ethoxy}ethoxy)ethoxy]ethoxy}piperidine-1-carboxylate (**7**)

A mixture of *tert*-butyl 4-hydroxypiperidine-1-carboxylate (0.50 g, 2.49 mmol) and NaH (0.3 g, 60%, 7.47 mmol) in THF (35.0 mL) was stirred at 0 °C for 1 h. Then, compound **6** (1.10 g, 2.55 mmol) was slowly added to the reaction, which was stirred at room temperature overnight. The mixture was diluted with water (30 mL) and extracted with EtOAc (30 mL × 3). The combined organic phase was washed with brine, dried over anhydrous Na_2_SO_4_, and evaporated. The residue was purified by column chromatography on silica gel [EtOAc-petroleum ether (30–70%)] to give the product (0.58 g, 51%) of colorless oil. HPLC analysis was as follows: 96.8%; ^1^H-NMR (400 MHz, CDCl_3_) δ 4.63 (t, *J* = 3.2 Hz, 1H), 3.90–3.84 (m, 2H), 3.79–3.76 (m, 2H), 3.69–3.63 (m, 15H), 3.51–3.45 (m, 2H), 3.09–3.02 (m, 2H), 1.85–1.81 (m, 3H), 1.76–1.68 (m, 1H), 1.63–1.49 (m, 6H), and 1.45 (s, 9H); ^13^C-NMR (CDCl_3_, 100 MHz) δ 154.8, 98.9, 79.4, 75.1, 70.8, 70.7, 70.6, 70.5, 67.4, 66.6, 62.2, 31.0, 30.6, 28.4, 25.4, and 19.5.

#### 3.1.4. *N*-{(*S*)-1-[4-(2-{2-[2-(2-mesyloxyethoxy)ethoxy]ethoxy}ethoxy)piperidin-1-yl]-3-(4-fluorophenyl)-1-oxopropan-2-yl}-5-chloro-1H-indole-2-carboxamide (**12**)

Under an ice bath, methanesulfonyl chloride (0.02 mL, 0.25 mmol) was slowly added in groups to a mixture of compound **11** (0.15 g, 0.24 mmol) and Et_3_N (0.10 mL, 0.72 mmol) in anhydrous CH_2_Cl_2_ (25 mL). The reaction mixture was stirred at room temperature for 5 h. Then, the reaction mixture was diluted with CH_2_Cl_2_ and washed with water, 1 M aqueous HCl, and saturated NaHCO_3_ and brine, and then dried and evaporated over anhydrous Na_2_SO_4_. The residue was purified by column chromatography on silica gel [CH_3_OH-CH_2_Cl_2_ (0–2%)] to give the product (0.15 g, 89%) as a white solid. HPLC analysis was as follows: 100.0%; m.p. 130–132 °C. ^1^H-NMR (400 MHz, CDCl_3_) δ 9.64 (d, *J* = 15.2 Hz, 1H), 7.58 (s, 1H), 7.52 (t, *J* = 6.8 Hz, 1H), 7.31 (t, *J* = 6.0 Hz, 1H), 7.21 (d, *J* = 6.8 Hz, 1H), 7.16–7.14 (m, 2H), 6.95 (t, *J* = 6.8 Hz, 2H), 6.91–6.86 (m, 1H), 5.37–5.33 (m, 1H), 4.38–4.35 (m, 2H), 3.84–3.74 (m, 3H), 3.66–3.59 (m, 14H), 3.49–3.25 (m, 2H), 3.15–3.05 (m, 5H), 1.83–1.50 (m, 3H), and 1.34–1.29 (m, 1H); ^13^C-NMR (CDCl_3_, 100 MHz) δ 169.2, 162.0 (d, *J* = 195.0 Hz), 160.5, 134.7, 131.8 (d, *J* = 15.4 Hz), 131.5, 131.1, 131.0, 130.9, 128.6, 126.2, 125.0, 121.2, 115.5 (d, *J* = 3.4 Hz), 115.3 (d, *J* = 3.6 Hz), 113.0 (d, *J* = 3.1 Hz), 73.5, 70.82, 70.76, 70.68, 70.5, 69.2, 69.0, 67.6, 50.0 (d, *J* = 7.7 Hz), 42.7, 39.3, 38.76, 38.47, 31.0 (d, *J* = 7.6 Hz), and 30.4 (d, *J* = 3.6 Hz); EIMS *m*/*z* 698.1 [M]^+^.

#### 3.1.5. General Procedure for Synthesis of Compounds **8** and **10**

Aqueous HCl (3 M, 2 mL) was slowly added to a solution of compound **7** or **9** (0.58 g, 1.26 mmol) in methanol (15 mL) in an ice bath. The reaction mixture was stirred at room temperature for 6 h, and then concentrated to give the crude target products **8** or **10**, which were used for the next reaction without further purification.

#### 3.1.6. General Procedure for Synthesis of Compounds **9** and **11**

HATU (1.0 equiv., dissolved in 1.5 mL DMF) and DIPEA (3.0 equiv., dissolved in 1.5 mL DMF) were added to a solution of *N*-(*tert*-Butoxycarbonyl)-4-fluoro-l-phenylalanine or 5-chloroindole-2-carboxylicacid (1.0 equiv.) in anhydrous DMF (1.5 mL). The reaction mixture was stirred at room temperature for 10 min. Then, compound **8** or **10** (1.0 equiv.) was added to the reaction. The mixture was stirred at 45 °C for 5 h. After cooling to room temperature, the mixture was diluted with water (30 mL) and extracted with EtOAc (30 mL × 3). The combined organic phase was washed with brine (1 L × 2), dried over anhydrous Na_2_SO_4_, and evaporated. The residue was purified by reversed-phase chromatography (methanol-water (20–90%)) to give the product **9** or **11**.

*(S)-3-(4-Fluorophenyl)-1-{4-[2-(2-{2-[2-(tetrahydro-2H-pyran-2-yloxy)ethoxy]ethoxy}ethoxy)ethoxy]piperidin-1-yl}-2-(tert-butoxycarbonylamino)propan-1-one* (**9**). HPLC analysis was as follows: 98.9%; ^1^H-NMR (400 MHz, CDCl_3_) δ 7.16–7.12 (m, 2H), 6.98–6.94 (m, 2H), 5.39 (t, *J* = 6.4 Hz, 1H), 4.84–4.79 (m, 1H), 3.81–3.71 (m, 3H), 3.67–3.58 (m, 14H), 3.54–3.18 (m, 4H), 2.98–2.92 (m, 2H), 1.79–1.48 (m, 3H), 1.41 (d, *J* = 3.6 Hz, 9H), and 1.35–1.26 (m, 1H); ^13^C-NMR (CDCl_3_, 100 MHz) δ 169.6, 161.9 (d, *J* = 194 Hz), 155.0, 132.3 (d, *J* = 18.4 Hz), 131.0 (d, *J* = 5.9 Hz), 130.9 (d, *J* = 6.0 Hz), 115.3 (d, *J* = 5.3 Hz), 115.1 (d, *J* = 5.2 Hz), 79.7, 73.7 (d, *J* = 5.2 Hz), 72.5, 70.81, 70.77, 70.65, 70.61, 70.3, 67.5, 61.7, 50.9, 42.6, 39.4, 39.1, 31.1(d, *J* = 3.8 Hz), 30.4, and 28.3.

*N-{(S)-1-[4-(2-{2-[2-(2-Hydroxyethoxy)ethoxy]ethoxy}ethoxy)piperidin-1-yl]-3-(4-fluorophenyl)-1-oxopropan-2-yl}-5-chloro-1H-indole-2-carboxamide* (**11**). HPLC analysis was as follows: 94.5%; m.p. 103–105 °C. ^1^H-NMR (400 MHz, CDCl_3_) δ 11.70 (t, *J* = 8.4 Hz, 1H), 8.91–8.86 (m, 1H), 7.70 (s, 1H), 7.40 (d, *J* = 6.8 Hz, 1H), 7.34 (dd, *J* = 6.4, 4.4 Hz, 2H), 7.25 (s, 1H), 7.17 (dd, *J* = 6.8, 1.6 Hz, 1H), 7.05 (t, *J* = 6.8 Hz, 2H), 5.15–5.10 (m, 1H), 4.55–4.49 (m, 1H), 3.98–3.79 (m, 1H), 3.73–3.65 (m, 1H), 3.52–3.48 (m, 14H), 3.42–3.39 (m, 2H), 3.33–3.21 (m, 1H), 3.14–2.94 (m, 3H), 1.75–1.66 (m, 2H), and 1.39–1.19 (m, 2H); ^13^C-NMR (CDCl_3_, 100 MHz) δ 169.4, 161.5 (d, *J* = 190.3 Hz), 160.6, 135.3, 134.4 (d, *J* = 7.5 Hz), 132.9, 131.7, 131.6, 128.5, 124.7, 124.0, 121.1, 115.3, 115.2, 114.3, 103.4, 74.6, 73.8, 72.8, 70.5, 70.3, 70.2, 67.2 (d, *J* = 6.1 Hz), 60.7, 50.6 (d, *J* = 9.6 Hz), 43.1, 42.7, 37.0 (d, *J* = 8.8 Hz), 31.7 (d, *J* = 33.1 Hz), and 31.1 (d, *J* = 41.3 Hz). EIMS *m*/*z* 620.1 [M]^+^.

#### 3.1.7. General Procedure for Synthesis of Compounds **3** and **13**

NaN_3_ (3.0 equiv.) was added to a solution of compound **12** or **20** (1.0 equiv.) in anhydrous DMF (10 mL). The reaction mixture was stirred at 60 °C overnight. Then, the reaction mixture was poured into ice water and extracted with EtOAc (30 mL × 3). The combined organic phase was washed with brine and dried over anhydrous Na_2_SO_4_. The residue was purified by column chromatography on silica gel [CH_3_OH-CH_2_Cl_2_ (0–2%)] to give the product (0.10 g, 70%) as a thick solid.

*N-{(S)-1-[4-(2-{2-[2-(2-Azidoethoxy)ethoxy]ethoxy}ethoxy)piperidin-1-yl]-3-(4-fluorophenyl)-1-oxopropan-2-yl}-5-chloro-1H-indole-2-carboxamide* (**13**). HPLC analysis was as follows: 96.7%; m.p. 93–95 °C; ^1^H-NMR (400 MHz, *d*_6_-DMSO) δ 11.71 (d, *J* = 6.8 Hz, 1H), 8.91–8.86 (m, 1H), 7.70 (s, 1H), 7.33 (t, *J* = 5.6 Hz, 2H), 7.25(s, 1H), 7.17 (d, *J* =7.2 Hz, 1H), 7.05 (t, *J* = 6.8 Hz, 2H), 5.17–5.09 (m, 1H), 3.98–3.79 (m, 1H), 3.73–3.65 (m, 1H), 3.59–3.48 (m, 15H), 3.41–3.34 (m, 2H), 3.25–2.94 (m, 4H), 1.78–1.65 (m, 2H), and 1.39–1.20 (m, 2H); ^13^C-NMR (*d*_6_-DMSO, 100 MHz) δ 169.3, 161.4 (d, *J* = 192.1 Hz), 160.6, 135.3, 134.4, 132.9, 131.7, 131.6, 128.5, 124.7, 124.0, 121.1, 115.3 (d, *J* = 2.5 Hz), 115.1 (d, *J* = 2.6 Hz), 114.3, 103.4 (d, *J* = 3.9 Hz), 74.6, 73.8, 70.5, 70.3, 70.2, 70.1, 70.0, 67.2 (d, *J* = 6.4 Hz), 50.6 (d, *J* = 8.4 Hz), 50.5, 43.1, 42.7, 37.0 (d, *J* = 6.8 Hz), 31.7 (d, *J* = 33 Hz), and 31.1 (d, *J* = 40.1 Hz). EIMS *m*/*z* 644.8 [M]^+^.

*N-[(S)-1-(4-{2-[2-(2-{2-[4-({[(S)-2-(4-Benzoylbenzamido)-6-(2-azidoacetamido)hexanoyl]oxy}methyl)-1H-1,2,3-triazol-1-yl]ethoxy}ethoxy)ethoxy]ethoxy}piperidin-1-yl)-3-(4-fluorophenyl)-1-oxopropan-2-yl]-5-chloro-1H-indole-2-carboxamide* (**3**). HPLC analysis was as follows: 98.8%; m.p. 113–115 °C; ^1^H-NMR (400 MHz, *d*_6_-DMSO) δ 11.74 (d, *J* = 9.6 Hz, 1H), 8.97–8.91 (m, 2H), 8.13 (d, *J* = 5.2 Hz, 1H), 8.10 (t, *J* = 5.6 Hz, 1H), 8.05 (d, *J* = 8.0 Hz, 2H), 7.83 (d, *J* = 8.0 Hz, 2H), 7.77 (d, *J* = 7.6 Hz, 2H), 7.71 (t, *J* = 6.8 Hz, 2H), 7.59 (t, *J* = 7.6 Hz, 2H), 7.40 (d, *J* = 8.8 Hz, 1H), 7.34 (t, *J* = 6.8 Hz, 2H), 7.26 (s, 1H), 7.18 (d, *J* = 8.4 Hz, 1H), 7.06 (t, *J* = 8.8 Hz, 2H), 5.21 (s, 2H), 5.14–5.07 (m, 1H), 4.58–4.51 (m, 2H), 4.49–4.43 (m, 1H), 4.00–3.97 (m, 1H), 3.83–3.79 (m, 4H), 3.73–3.66 (m, 1H), 3.49–3.45 (m, 13H), 3.31–3.17 (m, 1H), 3.13–2.91 (m, 5H), 1.83–1.65 (m, 4H), and 1.43–1.23 (m, 6H); ^13^C-NMR (CDCl_3_, 100 MHz) δ 195.9, 172.3, 169.4, 167.5, 166.5, 161.4 (d, *J* = 238.3 Hz), 160.6, 142.0, 139.9, 137.5, 137.1, 135.3, 134.5, 133.5, 132.9, 131.7, 131.6, 130.2, 129.9, 129.1, 128.5, 128.2, 125.6, 124.7, 124.0, 121.1, 115.3, 115.1, 114.3, 103.3, 74.6, 73.9, 70.5, 70.2 (d, *J* = 6.8 Hz), 70.1 (d, *J* = 7.9 Hz), 70.0 (d, *J* = 6.4 Hz), 69.1, 67.2 (d, *J* = 6.9 Hz), 58.3, 53.3, 51.3, 50.6 (d, *J* = 9.4 Hz), 49.9, 43.1, 42.7, 38.8, 36.9 (d, *J* = 8.8 Hz), 31.7 (d, *J* = 34.9 Hz), 31.1 (d, *J* = 42.8 Hz), 30.5, 29.0, and 23.6. EIMS *m*/*z* 1121.0 [M]^+^.

#### 3.1.8. General Procedure for Synthesis of Compounds **1**, **2,** and **20**

CuSO_4_·5H_2_O (0.19 equiv.) and Vc-Na (0.4 equiv.) were added to a mixture of compound **13** (1.0 equiv.) and (S)-prop-2-ynyl 2-(4-benzoylbenzamido)-6-(2-chloroacetamido)hexanoate (1.2 equiv.) in CH_2_Cl_2_/H_2_O (4 mL, *v*:*v* = 1:1). The reaction mixture was stirred at room temperature overnight. Then, the reaction mixture was diluted with EtOAc and washed with water, dried over anhydrous Na_2_SO_4_, and evaporated. The residue was purified by HPLC to give the product as a white solid.

*N-[(S)-1-(4-{2-[2-(2-{2-[4-({[(S)-2-(4-Benzoylbenzamido)-6-(2-chloro-acetylamino)hexanoyl]oxy}methyl)-1H-1,2,3-triazol-1-yl]ethoxy}ethoxy)ethoxy]ethoxy}piperidin-1-yl)-3-(4-fluorophenyl)-1-oxopropan-2-yl]-5-chloro-1H-indole-2-carboxamide* (**20**). HPLC analysis was as follows: 95.9%; m.p. 137–139 °C; ^1^H-NMR (400 MHz, *d*_6_-DMSO) δ 11.74 (d, *J* = 9.6 Hz, 1H), 8.97–8.91 (m, 2H), 8.22 (t, *J* = 5.2 Hz, 1H), 8.13 (d, *J* = 5.6 Hz, 1H), 8.04 (d, *J* = 8.0 Hz, 2H), 7.82 (d, *J* = 8.4 Hz, 2H), 7.77 (d, *J* = 7.2 Hz, 2H), 7.71 (t, *J* = 7.6 Hz, 2H), 7.59 (t, *J* = 7.6 Hz, 2H), 7.39 (d, *J* = 8.8 Hz, 1H), 7.34 (t, *J* = 6.8 Hz, 2H), 7.26 (s, 1H), 7.17 (d, *J* = 8.8 Hz, 1H), 7.06 (t, *J* = 8.8 Hz, 2H), 5.20 (s, 2H), 5.15–5.09 (m, 1H), 4.55–4.51 (m, 2H), 4.47–4.42 (m, 1H), 4.02 (s, 2H), 4.00–3.96 (m, 1H), 3.82–3.78 (m, 2H), 3.73–3.65 (m, 1H), 3.49–3.43 (m, 13H), 3.30–3.21 (m, 1H), 3.10–2.93 (m, 5H), 1.84–1.65 (m, 4H), and 1.48–1.23 (m, 6H); ^13^C-NMR (*d*_6_-DMSO, 100 MHz) δ 195.9, 172.3, 169.4, 166.5, 166.2, 161.5 (d, *J* = 238.3 Hz), 160.6, 142.0, 139.9, 137.5, 137.1, 135.3, 134.5, 133.5, 132.9, 131.7, 131.6, 130.2, 129.9, 129.1, 128.5, 128.2, 125.6, 124.7, 124.0, 121.1, 115.4, 115.1, 114.3, 103.3, 74.6, 73.9, 70.5, 70.2 (d, *J* = 6.1 Hz), 70.0 (d, *J* = 3.4 Hz), 69.9 (d, *J* = 1.7 Hz), 69.1, 67.1 (d, *J* =7.4 Hz), 58.2, 53.3, 50.7, 49.9, 43.1, 42.7, 39.1, 36.9 (d, *J* = 8.1 Hz), 31.7 (d, *J* = 36.1 Hz), 31.1(d, *J* = 43.4 Hz), 30.4, 28.9, and 23.5. EIMS *m*/*z* 1114.0 [M]^+^.

*N-[(S)-1-(4-{2-[2-(2-{2-[4-({[(S)-2-(4-Benzoylbenzamido)-6-(dansylamide)hexanoyl]oxy}methyl)-1H-1,2,3-triazol-1-yl]ethoxy}ethoxy)ethoxy]ethoxy}piperidin-1-yl)-3-(4-fluorophenyl)-1-oxopropan-2-yl]-5-chloro-1H-indole-2-carboxamide* (**1**). HPLC analysis was as follows: 100.0%; m.p. 151–153 °C; ^1^H-NMR (400 MHz, CD_3_OD) δ 8.73 (d, *J* = 10.0 Hz, 1H), 8.57 (d, *J* = 10.8 Hz, 1H), 8.41 (d, *J* = 11.2 Hz, 1H), 8.23 (d, *J* = 9.6 Hz, 1H), 8.10 (d, *J* = 12.0 Hz, 1H), 8.04–8.01 (m, 2H), 7.88–7.82 (m, 4H), 7.71 (t, *J* = 11.2 Hz, 1H), 7.62–7.55 (m, 5H), 7.44 (d, *J* = 11.6 Hz, 1H), 7.33–7.20 (m, 4H), 7.13 (s, 1H), 7.02 (t, *J* = 11.2 Hz, 2H), 5.38–5.33 (m, 3H), 4.81–4.76 (m, 1H), 4.62–4.58 (m, 3H), 3.93–3.81 (m, 4H), 3.63–3.59 (m, 13H), 3.24–3.07 (m, 3H), 2.99–2.92 (m, 8H), 1.90–1.72 (m, 4H), and 1.61–1.40 (m, 6H); ^13^C-NMR (*d*_6_-DMSO, 100 MHz) δ 195.9, 172.2, 169.3, 166.5, 162.7, 160.6, 158.7 (d, *J* = 30 Hz), 151.8, 142.0, 139.9, 137.4, 137.1, 136.6, 135.3, 133.5, 132.9, 131.7, 131.6, 130.2, 129.9, 129.8, 129.5, 129.4, 129.1, 128.6, 128.5, 128.2, 128.1, 125.6, 124.7, 124.0, 121.1, 119.6, 115.5, 115.3, 115.1, 114.3, 103.4, 74.6, 73.8, 70.4, 70.2 (d, *J* = 5.8 Hz), 70.1 (d, *J* = 8.0 Hz), 70.0 (d, *J* = 5.1 Hz), 69.1, 67.1 (d, *J* = 7.0 Hz), 58.2, 53.3, 49.9, 49.0, 45.5, 42.6, 30.2, 29.5, 29.2, 27.0, and 23.2. EIMS *m*/*z* 1271.0 [M]^+^.

*N-[(S)-1-(4-{2-[2-(2-{2-[4-({[(S)-2-[(S)-2-(4-Benzoylbenzamido)-6-(dansylamide)hexanamido]-6-(5-{(3aR,4S,6aS)-2-oxohexahydro-1H-thieno[3,4-d]imidazol-1-yl}pentanamido)hexanoyl]oxy}methyl)-1H-1,2,3-triazol-1-yl]ethoxy}ethoxy)ethoxy]ethoxy}piperidin-1-yl)-3-(4-fluorophenyl)-1-oxopropan-2-yl]-5-chloro-1H-indole-2-carboxamide* (**2**). HPLC analysis was as follows: 99.4%; m.p. 159–161 °C; ^1^H-NMR (400 MHz, *d*_6_-DMSO) δ 11.74 (d, *J* = 10.0 Hz, 1H), 8.98–8.92 (m, 1H), 8.58 (d, *J* = 7.6 Hz, 1H), 8.44 (d, *J* = 8.8 Hz, 1H), 8.34 (d, *J* = 7.2 Hz, 1H), 8.30 (d, *J* = 8.8 Hz, 1H), 8.11–8.09 (m, 2H), 8.02 (d, *J* = 8.0 Hz, 2H), 7.91 (t, *J* = 5.6 Hz, 1H), 7.81–7.69 (m, 7H), 7.63–7.54 (m, 4H), 7.39 (dd, *J* = 8.8, 1.6 Hz, 1H), 7.34 (t, *J* = 6.4 Hz, 2H), 7.26 (s, 1H), 7.23 (d, *J* = 7.6 Hz, 1H), 7.17 (d, *J* = 8.8 Hz, 1H), 7.06 (t, *J* = 8.8 Hz, 2H), 6.43 (br s, 1H), 6.37 (br s, 1H), 5.18–5.11 (m, 3H), 4.53–4.49 (m, 2H), 4.45–4.40 (m, 1H), 4.28 (t, *J* = 6.4 Hz, 1H), 4.24–4.21 (m, 1H), 4.11–4.08 (m, 1H), 4.00–3.97 (m, 1H), 3.82–3.77 (m, 2H), 3.72–3.64 (m, 1H), 3.50–3.44 (m, 13H), 3.30–3.18 (m, 1H), 3.13–2.90 (m, 6H), 2.81 (s, 6H), 2.78–2.76 (m, 3H), 2.56 (d, *J* = 12.4 Hz, 1H), 2.03 (t, *J* = 7.2 Hz, 2H), 1.73–1.56 (m, 6H), and 1.48–1.24 (m, 16H); ^13^C-NMR (*d*_6_-DMSO, 100 MHz) δ 195.9, 172.5, 172.3, 172.2, 169.4, 166.2, 163.2, 161.5 (d, *J* = 241.1 Hz), 160.6, 151.8, 141.9, 139.7, 137.9, 137.2, 136.6, 135.3, 135.1, 134.4, 133.5, 132.9, 131.7, 131.6, 130.2, 129.8, 129.6, 129.5, 129.1, 128.6, 128.5, 128.2, 128.1, 125.5, 124.7, 124.0, 121.1, 119.6, 115.5, 115.3, 115.1, 114.3, 103.4, 74.6, 73.9, 70.5, 70.2 (d, *J* = 6.2 Hz), 70.1 (d, *J* = 3.4 Hz), 70.0 (d, *J* = 1.7 Hz), 69.1, 67.1 (d, *J* = 7.1 Hz), 61.5, 59.7, 58.1, 55.9, 53.6, 52.4, 50.5, 49.9, 45.5, 43.1, 42.9, 38.6, 35.7, 31.9, 31.4 (d, *J* = 6.2 Hz), 30.9 (d, *J* = 10.9 Hz), 29.6, 29.2, 28.7, 28.5,25.8, 23.3, and 23.2. EIMS *m*/*z* 1625.0 [M]^+^.

### 3.2. Enzyme Kinetics

The inhibitory activity of the test compounds against RMGPa was monitored using a microplate reader (BIO-RAD, Bio-Rad Laboratories, Hercules, CA, USA), based on the published method [16]. In brief, RMGPa activity was measured in the direction of glycogen synthesis by the release of phosphate from glucose-1-phosphate. Each test compound was dissolved in DMSO and diluted at different concentrations for IC_50_ determination. The enzyme was added to 100 μL of buffer containing 50 mM Hepes (pH = 7.2), 100 mM KCl, 2.5 mM MgCl_2_, 0.5 mM glucose-1-phosphate, 1 mg/mL glycogen, and the test compound in 96-well microplates (Costar, Corning Incorporated, Corning, NY, USA). After the addition of 150 μL of 1 M HCl containing 10 mg/mL ammonium molybdate and 0.38 mg/mL malachite green, reactions were run at 22 °C for 25 min, and then the phosphate absorbance was measured at 655 nm. The IC_50_ values were estimated by fitting the inhibition data to a dose-dependent curve using a logistic derivative equation.

### 3.3. Production of Acute Myocardial Infarction

Male Sprague–Dawley rats weighing 250–350 g were intraperitoneally anesthetized with sodium phenobarbital (10 μL/g). The rats were intubated and ventilated with a volume-cycled small-animal ventilator (Chengdu Taimeng Science And Technology Co. Ltd., Chengdu, China). An anterior thoracotomy was performed to open the pericardium. The heart was then rapidly exteriorized, and a 6-0 silk suture was tightened around the proximal left anterior descending coronary artery (before the first branch of diagonal artery). Positive end-expiratory pressure was applied to fully inflate the lungs. The muscle layer and skin were closed separately after plasmid injection, and the animals were allowed to recover. The experimental protocol was approved by the Chengde Medical University committee on animal experiments.

### 3.4. Photoaffinity Labeling Experiment

Blood samples prepared from rats with acute myocardial infarction were drawn from the abdominal aorta in rats after coronary ligation using a 23-gauge needle. The serum blank was obtained from male rats without surgery. The blood sample was stored at 4 °C for 30 min, and then centrifuged at 3000 rev./min for 15 min at 4 °C. Serum was frozen at −80 °C until assay. The protein concentration of the suspension was measured according to the Bradford method. The serum proteomes were diluted to 2.0 mg/mL with 50 mM Tris·HCl buffer (pH 7.4). The labeling reaction was initiated by incubating proteomes with the probe (dissolved in DMSO) at 4 °C for 8 h, and then exposed to UV 365 nm (220 v, 6 W, 365 nm) at a distance of 3 cm. The reaction mixture was centrifuged at 48,000× *g* for 10 min, then the supernatant was removed and the precipitate was resuspended in lysis buffer (urea 480 mg/L, chaps 40 mg/L, Tris-base 4.8 mg/L, DTT 10 mg/L, Ampholate 50 mL/L, and bromophenol blue 0.002%) at 4 °C for 1 h, and then centrifuged at 18,000× *g* for 2 h. The supernatant was dialyzed against 50 mM Tris·HCl buffer (pH 7.4). To detect the proteins photo-cross-linked by test probe, the homogenate photolabeled with the probe was subjected to SDS-PAGE electrophoresis, and then transferred onto a polyvinylidene fluoride membrane. The membrane was blocked with 1% BSA in PBS with 0.05% Tween 20 at room temperature for 1 h, washed twice using PBS with 0.05% Tween 20 for 5 min, and then imaged using the Tanon-6100 Chemiluminescent Imaging system (Tanon Science and technology Co., Ltd., Shanghai, China). The band densities were calculated with Quantity One software 4.6.2 (Bio-Rad Laboratories, Inc., Hercules, CA, USA). The serum in the rat acute myocardial infarction model treated with probe 1 (Figure 4, lane 2) was excised in gel digested with trypsin, and subjected to LC-MS/MS. The data were recorded on an LTQ-Orbitrap Fusion mass spectrometer (ThermoFisher Scientific Inc., Waltham, MA, USA) coupled to an Easy-nLC 1000 LC System (Thermo Fischer Scientic). Label-free MS analysis was performed by Thermo Q-Exactive mass spectrometry (ThermoFisher Scientific Inc., Waltham, MA, USA), and the MS raw data were processed using MaxQuant 1.5.2.8 software with the Uniprot-Orytolagus cuniculus database.

## 4. Conclusions

In summary, three novel activity-based probes were designed, synthesized, and applied in the semiquantitative analysis of serum-free-GP levels after an acute myocardial infarction. The synthesized probes were evaluated for their potency in an enzyme inhibition assay against RMGPa. The results showed that the probe **1** exhibited inhibitory activity against RMGPa with an IC_50_ value of 13.20 μM, which was approximately 100-fold less potent than that of CP-320626. Photoaffinity labeling studies were also performed, and the protein with a MW of about 90–100 kDa was efficiently and specifically labeled by probe **1**. This was consistent with the MW of GP. The attempt to semi-quantitatively evaluate the GP level in serum with probe **1** was also performed, and a protein band with a MW of about 90–100 kDa was detected. Mass spectrometric analysis revealed that the GPMM was well-preserved in the bands. For semiquantitative data analysis, the concentration of the GPMM protein was found to be between 25 and 50 ng/mL.

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
