# Peer review of "Design, Synthesis, and Use of Novel Photoaffinity Probes in Measuring the Serum Concentration of Glycogen Phosphorylase"

_molecules, 2019, doi:10.3390/molecules24040798_

Reviewer 1 Report

Manuscript ID molecules-392821

Title

Design, Synthesis and Use of Novel Photoaffinity Probes in Attempt to Measure the Serum Concentration of Glycogen Phosphorylase

Authors

Zhang Yuchao , Wang Youde , Yan Zhiwei , Song Chengjun , Miao Guangxin , Zhang Liying *

Review: This paper conducts the synthesis of three benzophenone photoaffinity probes with different secondary tags. These probes are based on elaboration of the bioactive motif CP-320626 a potent inhibitor of glycogen phosphorylase. The authors then attempt to use this in measuring serum concentrations of the enzyme via semi-quantitative protein electrophoretic shift. A protein band which may contain the enzyme of interest was identified and evaluated using mass spectrometry. The aim was to give early detection method for ischaemia.

Spelling and grammar require attention alongwith spacings between numbers and units and variations in significant figures. The whole document needs to be checked carefully. By way of example but by no means exhaustive:

 Intro Line 4 “human”

“Clinical diagnosis of GP require rapid and reliable test systems” GP is not a diagnosis it is an enzyme.

“This is because most of the studies covered only for measurement of”

“avoid the sterically hindrance caused” should be steric hindrance

“3M HCl” should have a space between number and units.

“69 % yield” should not have a space before a %

“Similarly as probe 1” should be: similar to probe 1

“A procedure attempt to semiquantitative analysis” should be: An attempt to semiquantitatively analyse…

Statement on P2 Para 2 that “biotin is not optimal …due to the time” time for what – this needs to be explained further.

The chemistry is well explained just attention to typographical errors is required.

It is pleasing that the authors actually attempt to use these probes and do not just synthesise them.

Given the azide resulted in greatly diminished activity in the assay the equivalent alkyne should be investigated as the azide could be incorporated onto the reporter unit for subsequent click chemistry.

“MW about 90-100 kDa, was clearly labelled, and thus most likely the MW of GP” it either is, or is not the weight of GP – is this not known? This statement is repeated through the document. Perhaps “consistent with the MW of GP” would be more appropriate.

In section 2.2 why was the 10 µM concentration chosen for the probe – this was not a concentration tested in the gel (Fig 3).

“linear relationship was obtained between optical density and the specific plasma protein concentration” The optical density values are not included/graphed here only the picture of the gel. It is hard to say from the picture of the gel that this is a linear relationship directly proportional to protein concentration. Also was this repeated to assess error.

I would be appropriate to have discussion around the mass spec results this is left to the reader to interpret. How many proteins are contained in the band highlighted? Is it just one? It would be much better to have the isolated protein band and microsequencing included in the paper to give more confidence in the results.

I would also like the authors to indicate the limiations of their current study and indicate what is next.     

Author Response

Dear Editor,

Thank you very much for your message. Our responses to the reviewers’ comments are as below:

Reviewer comments: Reviewer 1: This paper conducts the synthesis of three benzophenone photoaffinity probes with different secondary tags. These probes are based on elaboration of the bioactive motif CP-320626 a potent inhibitor of glycogen phosphorylase. The authors then attempt to use this in measuring serum concentrations of the enzyme via semi-quantitative protein electrophoretic shift. A protein band which may contain the enzyme of interest was identified and evaluated using mass spectrometry. The aim was to give early detection method for ischaemia.

Spelling and grammar require attention alongwith spacings between numbers and units and variations in significant figures. The whole document needs to be checked carefully. By way of example but by no means exhaustive:

Many thanks for this careful observation. We are sorry for our mistakes and carelessness. All the mistakes have been revised accordingly.

Intro Line 4 “human”

Many thanks for this careful observation. We are sorry for our mistakes and carelessness. The erroneous word " huamn " has been revised as " human " in our manuscript on Page 1, Line 35.

“Clinical diagnosis of GP require rapid and reliable test systems” GP is not a diagnosis it is an enzyme.

Many thanks for the reviewer’s suggestion. We have replaced “clinical diagnosis” with “clinical detection” on Page 2, Line 45.

“This is because most of the studies covered only for measurement of”

Many thanks for this careful observation. We have replaced “This is because most of the studies covered only for measurement of GPBB but not GPBB and GPMM efflux” with “This is because most of the studies covered only for measurement of GPBB but not the efflux of GPBB and GPMM” on Page 2, Line 50.

“Similarly as probe 1” should be: similar to probe 1

Many thanks for the reviewer’s suggestion. We have replaced “Similarly as” with “It is similar to” on Page 5, Line 112.

“A procedure attempt to semiquantitative analysis” should be: An attempt to semiquantitatively analyse…

Many thanks for this careful observation. We have replaced “A procedure attempt to semiquantitative analysis” with “An attempt to semiquantitatively analyze” on Page 1, Line 23, Page 7, Line 160 and Page 18, Line 424.

Statement on P2 Para 2 that “biotin is not optimal …due to the time” time for what – this needs to be explained further.

Many thanks for this suggestion. The sentence is explained in the revised manuscript On Page 2, Line 62, and the added sentences are as follows:" While a biotin tag is ideal for affinity tagging purposes, its use is not optimal for higher throughput activity-based profiling due to the time and labor involved in producing western blots. Additionally, many cells and tissues contain endogenously biotinylated proteins that complicate analysis of biotinylated probe-based western blots."

The chemistry is well explained just attention to typographical errors is required.

Many thanks for this careful observation. We are sorry for our mistakes and carelessness. All the mistakes have been revised accordingly on Page 3, Line 93, Page 4, Line 101, 102, 103, 106, 108, Page 5, Line 110, 114, 116 and Page 6, Line 119, 124.

It is pleasing that the authors actually attempt to use these probes and do not just synthesise them.

Thank you very much for your recognition of our work.

Given the azide resulted in greatly diminished activity in the assay the equivalent alkyne should be investigated as the azide could be incorporated onto the reporter unit for subsequent click chemistry.

Many thanks for the reviewer’s suggestion. We are now in the process of introducing the alkyne group onto the reporter unit for subsequent click chemistry, and we will report the results in due course.

“MW about 90-100 kDa, was clearly labelled, and thus most likely the MW of GP” it either is, or is not the weight of GP – is this not known? This statement is repeated through the document. Perhaps “consistent with the MW of GP” would be more appropriate.

Many thanks for this careful observation. We have replaced “MW about 90-100 kDa, was clearly labelled, and thus most likely the MW of GP” with “MW about 90-100 kDa, which is consistent with the MW of GP, was clearly labelled” on Page 1, Line 22, Page 7, Line 152, Page 8, Line 167 and Page 18, Line 423.

In section 2.2 why was the 10 µM concentration chosen for the probe – this was not a concentration tested in the gel (Fig 3).

Many thanks for the reviewer’s suggestion. The effects of different concentration of probe 1 (including the 10 µM concentration) on labeling efficiency was investigated. Our results show that the efficiency is relatively good when the concentration level of probe 1 in labeling experiment increases from 30 nM to 10 mM. However, since streptavidin blot analysis is not sufficiently specific, there are some "nonspecific" bands appeared on the blot in our studies when the concentration level increases from 10 µM to 10 mM. Therefore, the 10 µM concentration was chosen for the probe.

“linear relationship was obtained between optical density and the specific plasma protein concentration” The optical density values are not included/graphed here only the picture of the gel. It is hard to say from the picture of the gel that this is a linear relationship directly proportional to protein concentration. Also was this repeated to assess error.

Many thanks for this careful observation. We are sorry for our mistakes and. The sentence "linear relationship was obtained between optical density and the specific plasma protein concentration" has been deleted in our revised manuscript on Page 8, Line 168.

I would be appropriate to have discussion around the mass spec results this is left to the reader to interpret. How many proteins are contained in the band highlighted? Is it just one? It would be much better to have the isolated protein band and microsequencing included in the paper to give more confidence in the results.

Many thanks for this suggestion. It is true that more than one protein has been identified, and we have added the relevant results in the text On Page 8, Line 174. The added sentences are as follows:" Then we used a more detailed proteomic analysis, mass spectrometric analysis method, to detect the protein bands with MW about 90-100 kDa. Several proteins have been identified, and the results are summarized in Table 2. This analysis revealed that the GPMM was well preserved. The detailed results of LC-MS/MS analysis are in Figure 5."

I would also like the authors to indicate the limiations of their current study and indicate what is next.

The concentration of the target protein was successfully determined by using the affinity probe 1. The next experiment is to use other method to confirme and validate the plasma efflux of GPBB and GPMM. Since the probe 1 inhibited RMGPa with an IC50 value of 13.20 μM, being approximately 100-fold less potent than that of CP-320626. It makes the concentration of probe 1 to be high in the photoaffinity labeling experiment. In the future, the structure of probes should be further optimized in order to improve their activity. The molecular weight of probe 1 is large and the preparation is difficult, which is not suitable for industrialization. In the future, the synthesis process will be optimized to improve the yield. Due to the need for SDS-PAGE electrophoresis, the whole time period from blood collection to the results obtained is longer than that of immunoassay methods. It is necessary to further optimize the structure of probes in order to rapid detection.

Reviewer 2 Report

This paper describes development of photoaffinity probes for the semi-quantitative detection of serum GP concentration. Photoaffinity probes are designed appropriate for the purpose of detection, and synthetic procedures are concise and efficient. One of the three probes were found to be selectively detect GP, and which enabled determination of GP concentration in serum. I recommend publication of this paper in Molecules after consideration of points below.

1. The authors pointed that the steric interference is critical for binding of probes to the target protein. The probe 2 is bigger than the other probes, so it is likely to be less potent than the others. But, the probe 3 with azide moiety was found to be far less potent than the probe 1 with a dansyl group which still inhibits enzymatic activity of the target protein. Please explain.

2. The concentration of the target protein was successfully determined by using the affinity probe 1. However, it should be confirmed and validated by other method.

Author Response

Dear Editor,

Thank you very much for your message. Our responses to the reviewers’ comments are as below:

Reviewer 2:

1. The authors pointed that the steric interference is critical for binding of probes to the target protein. The probe 2 is bigger than the other probes, so it is likely to be less potent than the others. But, the probe 3 with azide moiety was found to be far less potent than the probe 1 with a dansyl group which still inhibits enzymatic activity of the target protein. Please explain.

Many thanks for this suggestion. More detailed analysis of the probe 3 has been described in the revised manuscript On Page 6, Line 136. There are many factors affecting the activity of probes. The presence of reactive group is a necessary but not sufficient condition for retaining the activity of probes. A large number of studies have reported that probes with similar molecular weight but slightly different chemical structures have different activities. For the probes involved in this study, the probe 3 exhibited no inhibitory activity against RMGPa in our repeated  experiments, therefore, we reported it truthfully in the paper. Due to the lack of binding mode of chemical probes with GP, it is impossible to estimate how the introduction of different functional elements affects the three-dimensional conformation of GP. We only confirm that there are significant differences in the regulation of active state of GP enzyme between probe 3 containing azide group and probe 1 without azide group. There are a large number of studies reporting how the azide functional group affect enzyme activity of compounds in Medicinal chemistry  [Bioorg. Med. Chem. Lett. 2013, 23, 6138-6140.; Bioorg. Med. Chem. 2000, 8, 2417-2425.]. We speculate that this difference of activity may be due to the interaction between the azide functional group of probe 3 and non-allosteric site of GP, thus affecting the allosteric inhibition of GP. This may bring new discovery for the studies of GP inhibitors, but it is not the focus of our study.

2. The concentration of the target protein was successfully determined by using the affinity probe 1. However, it should be confirmed and validated by other method.

Many thanks for this suggestion. However, it is difficult to determine the concentration of the target proteins by other method. This is because all of the reported studies, including immunoassay methods and chemical approaches, covered only for measurement of GPBB but not efflux of GPBB and GPMM. Therefore, the analysis of the efflux of GPBB and GPMM is interesting and no one has studied it before.

Round  2

Reviewer 1 Report

I am satisfied with the response to my review

Author Response

I am satisfied with the response to my review

Thank you very much for your recognition of our work.

We would like to thank the reviewers who took the time to review and provide comments on our paper.

Reviewer 2 Report

The authors' responses and revised version of manuscript regarding to effect of the strucutres of synthesized probes on their activities are satisfactory to this reviewer. And I understand it is difficult to determine the concentration of target protein by other methods due to the lack of validated method. Then, please provide calibration curve with error bars, at least, for the determination of protein concentration in the given sample. Although the authors claimed it is semiquantitative, it is important to give more confidence to the readers with validation of the developed method. I would recommend this paper for the publication in Molecules if the authors include calibration curve in the manuscript.

Author Response

1. The authors' responses and revised version of manuscript regarding to effect of the strucutres of synthesized probes on their activities are satisfactory to this reviewer. And I understand it is difficult to determine the concentration of target protein by other methods due to the lack of validated method. Then, please provide calibration curve with error bars, at least, for the determination of protein concentration in the given sample. Although the authors claimed it is semiquantitative, it is important to give more confidence to the readers with validation of the developed method. I would recommend this paper for the publication in Molecules if the authors include calibration curve in the manuscript.

Many thanks for this suggestion. We have added calibration curve and more detailed analysis in the revised manuscript in the text On Page 8, Line 172; Page 9, Line 189.

The added sentences are as follows:" Standard curve method was applied to detect the concentration of the protein's MW consistent with the MW of GP(Figure 4, Lane 2). Figure 5 shows the standard curve of the protein concentration. Corresponding to the standard curve, the concentration of the protein's MW consistent with the MW of GP was found to be between 25 and 50 ng/mL."

We would like to thank the reviewers who took the time to review and provide comments on our paper.